# Exploring the Phenomenon of the Additive Colour Process While Using a Computer Programme by 7–8-Year-Old Students

**Jan Amos Jelinek**

Institute of Human Development Support and Education, The Maria Grzegorzewska University, 02-353 Warsaw, Poland; jajelinek@aps.edu.pl

**Abstract:** The research described in this article concerns the understanding of the additive colour process by children aged seven to eight years (N = 24) and the effectiveness of learning about this phenomenon while using a computer-based multimedia educational programme (MEP). First the children's knowledge of the phenomenon was tested, then an intervention was organised for the experimental group during which they used the MEP, after which the children's knowledge was tested again. Based on an analysis of the children's conversations and their drawings, the way of understanding the phenomenon of additive colour was established. Three children's conceptions of the understanding of the effect of the additive colour phenomenon are described (e.g., confusion with pigment mixing (RGB = CMY) and the claim that the effect of combining two additive colour creates a third colour (R + G = B)). Children's behaviour during the use of the educational computer programme was also described and evaluated in terms of how close they were to the teaching strategy developed by the programme's authors. The partial effectiveness of the MPE for the use of conclusions in a paper-and-pencil test was also investigated.

**Keywords:** additive colour; concepts; computer-based learning; intervention; effectiveness; second-grade students; 7-year-old; 8-year-old

## 1. Introduction

The phenomenon of additive colour arises from the summation of light beams. This phenomenon is well described in the red, green and blue (RGB) light space. Combining the beams of these lights with maximum brightness at a single point results in white light, while the opposite is no light emission—darkness. This is because, along with the light beam, electromagnetic radiation waves are combined, which are read by the human sense of sight [1]. The combination of two RGB light beams results in one of the colours from the CMY colour space (cyan, magenta and yellow). Combining green and blue light produces blue (cyan to be precise), combining blue and red light produces a shade of violet (magenta), combining red and green light produces yellow light.

In Isaac Newton's (1704) colour wheel, the RGB and CMY colours in between complement each other [1,2]. The phenomenon of additive colour is naturally used by the human eye (Young's theory). The physiognomy of the human eye is now reproduced in the displays of electronic devices (e.g., TVs and smartphones), in which three coloured beams of light (RGB) of different intensities are combined to give the viewer the impression of derived colours. The difference between the phenomena of additive and subtractive colour was explained by Hermann von Helmholtz (1852). In the trichromatic theory (Young-Helmholtz), he concluded that the key to understanding this phenomenon is not the physical properties of light waves, but the perceptual properties of humans [1].

The phenomenon of additive colour was previously confused by Newton with a similar phenomenon of pigment mixing—subtractive colour [1]. This phenomenon occurs when the surface of the paper/ canvas is covered with successive layers of pigment. From

a physical perspective, the effect of the phenomenon is the subtraction of visible rays of different light lengths. Mixing blue and yellow pigment produces green, yellow and violet pigment produces red, and violet and sky-blue pigment produces blue (Figure 1a,b).

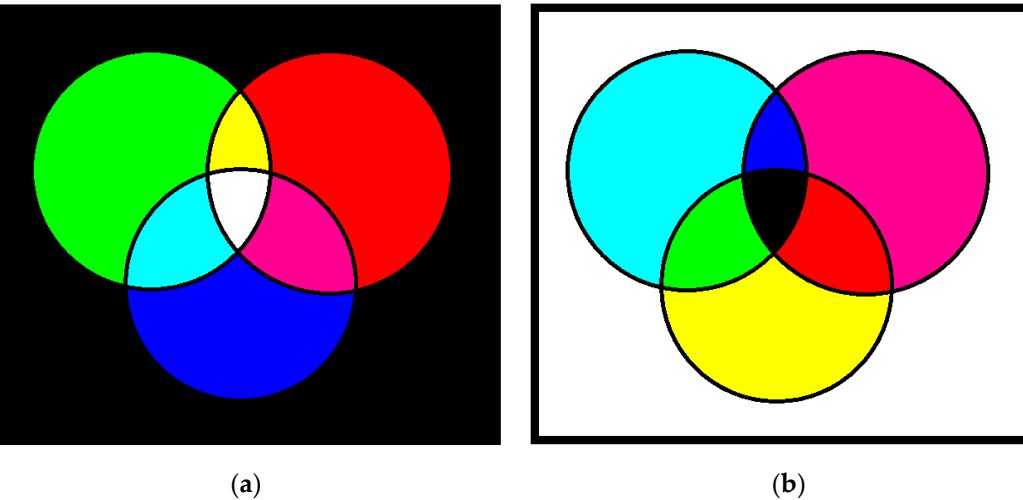

(**a**)　　　　　　　　　　　　　　　　　　　　　　　　(**b**)

**Figure 1.** (**a**) The difference between the blending of light colours in the phenomenon of additive colour. (**b**) The blending of paint pigments in subtractive colour.

In Polish schools, children have more opportunities to learn about the phenomenon of subtractive colour than additive colour. They are far more likely to gain experience with paint than with mixing coloured lights in a dark room. Science education teachers who propose to carry out the phenomenon of additive colour address it to children as young as preschool age [3]. It is assumed that by organising simple experiments in optics, the children's interest in cognition will be awakened and their cognitive development will be supported. However, the basis of support must be the knowledge of children's abilities and limitations related to the nature of children's cognitive development in making conclusions about optical phenomena [4–7]. In this context, research is needed to understand children's understanding of the nature of optical phenomena in order to organise effective support on this basis.

Research on children's understanding of light shows that, due its specificity, the concept of light is formed gradually in children. The development of the concept of light goes from attributing to it the properties of the objects from which it originates to gradually isolating these properties and treating light as a separate physical entity [2,5,8–10]. Research has shown that children have difficulty recognising the basic properties of light, and when asked about the effect of mixing colours of lights they talk about the effect of mixing pigments [2,4]. Children's explanations are thought to be influenced by their rich painting experiences in paint mixing [10,11].

Research shows that misconceptions about the properties of light are also manifested by teachers. They have poor knowledge of colour vision and confuse paint mixtures with mixtures of coloured lights, i.e., between additive and subtractive colour. Teachers hold a (more or less implicit) belief that colour is a feature of objects that is independent of both the type of light source illuminating the object and the properties of human vision [2,12,13].

Teachers' misconceptions may be the reason for forming false concepts in children about the properties of light and the phenomena associated with it. Because of these limitations, other forms of delivering science education are being sought. It is well known that the most effective way to teach physical phenomena is to learn about them directly, for example by conducting experiments. Children are able to understand basic information about the properties of light if the right conditions are provided and appropriate explanations are given [11,14–16]. Research shows that one possible form of support is the

use of computers in explaining the properties of light to children [17]. Research using educational computer games, which emphasises a playful form of child activity, is proving ineffective in teaching physical phenomena. The students who use them are not able to apply the knowledge they have acquired beyond the computer in paper-and-pencil tests [18]. However, multimedia experiential platforms, in which children can discover physical phenomena on their own by conducting experiments, may be more effective. In this study, one such programme was used to teach children the phenomenon of additive colour.

## 2. Materials and Methods

The aim of the research was to (a) determine how 7- and 8-year-old students understand the effect of mixing the three basic colours of visible light (RGB) and how they understand the formation of its effect and (b) whether they can apply the knowledge gained in the MEP in a paper-and-pencil test (beyond virtual reality).

The research is qualitative in nature. It was conducted in the second year of primary school. Twenty-four second-grade primary school students (7- and 8-year-olds), 15 boys and 9 girls, were included. The condition for selecting the group was (a) lack of visual problems, e.g., myopia and colour blindness (information obtained from the teacher) and (b) ignorance of the MEP programme used in the study: Socrates103. Fascynujące eksperymenty (published in Poland by Techland, 2012). This programme is designed for children from 7 to 12 years of age. The choice of programme was dictated by the prizes it had won in competitions dedicated to educational teaching resources. The programme resembles a multimedia experiential platform in which students discover physical phenomena by carrying out simple experiments (e.g., modifying the instrument components visible on the screen), and solving test tasks on their knowledge of each phenomenon (Figure 2).

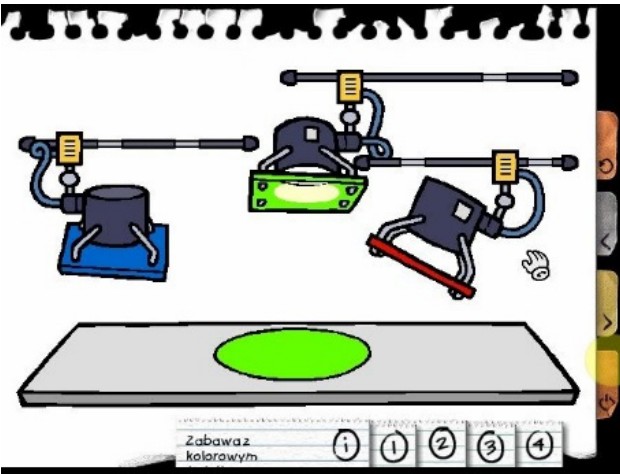

**Figure 2.** Screenshot of an additive colour task (Socrates103 programme).

The research was conducted using the pedagogical experiment method. First, a test was administered to check the students' knowledge, then a teacher intervention was introduced using a multimedia programme (MEP), and finally the same test was administered again to check students' knowledge.

The test checked students' knowledge of the 11 physical phenomena taught in the MEP. This test (a pre-test) took place in an individual form and was practical in nature. It consisted of demonstrating objects to the child and using them to carry out a fragment of the experiment, which allowed the children to get an idea of the phenomenon and the possibility of predicting its effect. For example, with regard to the phenomenon of additive colour, children were shown one torch (white light) and three foils (green, red and blue), as in Figure 3. The researcher would light the torch and then apply each foil

separately to the light source in turn. After such a demonstration, the researcher would stop and ask the child a question: What will be the colour of the light if I have two torches, put blue foil under one torch, and red foil under the second, and illuminate the same spot on the table with them? Similar questions were asked about other combinations of colour foil. The researcher handed the child a sheet of paper with schematically drawn torches and marked four combinations of light combinations (blue-red, green-red, green-blue and green-blue-red). There was a white rectangle under the torches, which he recommended to fill with with the appropriate colour, as in Figure 4. Each child was given a set of 12 crayons to draw colours with. The reason for choosing the drawing technique as the collection of children's responses was to avoid the need for children to name the colours and the possible difficulties arising from colour perception under test conditions (e.g., lighting).

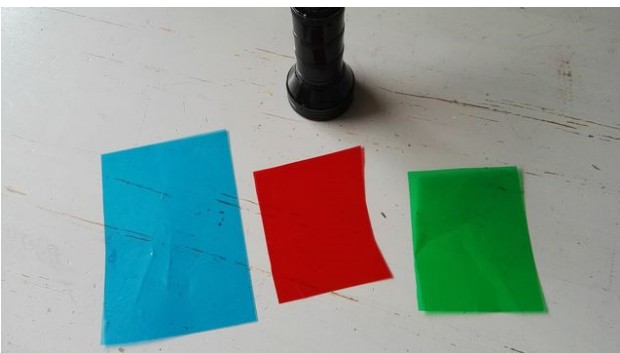

**Figure 3.** Photograph of the objects used in the study of understanding the phenomenon of additive colour.

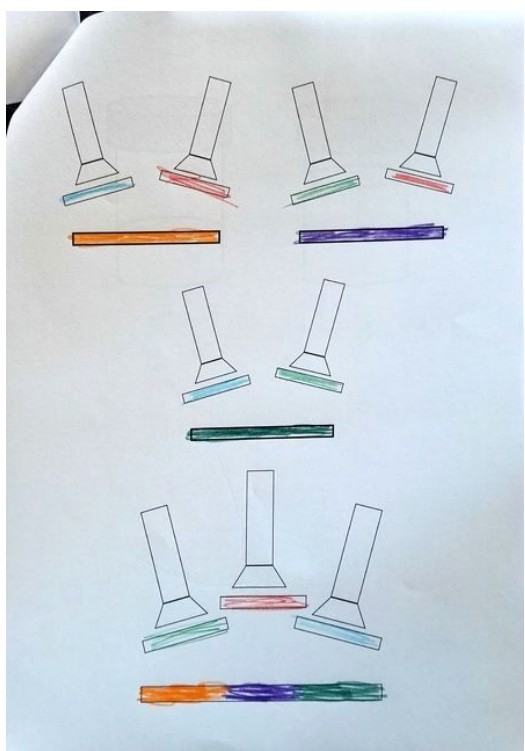

**Figure 4.** Example of a completed task by child—additive colour.

For the phenomenon of additive colour, on the computer screen the children saw lamps with three colours that they could move to illuminate a single point on the plane. In the test, the children saw a torch and three coloured foils (Figure 2).

Once the children solved the pre-test, the main phase of the experiment began. Using a randomised method, the children (24) were divided into two groups: experimental group (EG = 12) and control group (CG = 12). From then on, the CG children did not participate in the experiment until the post-test. The EG children participated in organised meetings (eight in total) with the MEP once a week for 8 weeks. These meetings were organised in randomly selected pairs in isolated rooms. Since leaving children alone in front of a computer screen proves to be ineffective [19], when organising the meetings, an effort was made to create conditions for the students to interact with each other without influencing their use of the MEP. This was done by (a) modifying the positioning of the computers on the tables in the room, and (b) introducing an element of competition by noting the points earned in the MEP for solving the tasks correctly. The meetings were as follows:

- The first meetings (1–3) were individual. Computers were set up back-to-back and the children used the MEP independently. No points were noted.
- In the two subsequent meetings (4–5), the layout of the computers remained the same, but the children were encouraged to compete by noting on the board the points they had scored in the MEP.
- The sixth meeting was held with the computers set up next to each other so that the children could observe each other's work on the screen.
- The seventh meeting was carried out by a pair of children on one computer (this encouraged more communication and information sharing).
- The final, eighth meeting, was conducted in a similar way to the first meetings—on separate computers, set up back-to-back.

In order to establish the process of children's learning about the phenomenon of additive colour, a recording programme was installed on the computers in addition to the MEP. It recorded screen images throughout the use of the MEP, while using a camera to record the students' behaviour in front of the computer. This resulted in 40 h of footage. During the analysis, the focus was on establishing the course of action that children take to learn about physical phenomena. It was determined whether children followed the teaching strategy of the authors of the MEP: whether they read the instructions for the experiments, whether and how they carried out the experiments by manipulating the elements visible on the screen, whether they marked the answer by trial and error when solving the tasks, and whether, when the programme revealed an error, they made further attempts to learn about the phenomenon or gave up looking for further attractions of the programme.

It should be added that the length of each meeting was not limited by time. Over the course of eight meetings, a shift from fascination with the new programme to boredom and reluctance was noticeable. It was the emotions accompanying the children when using the MEP that led to the experiment being completed earlier than planned (10 meetings). The average time children used the programme was 3.5 h. A post-test was conducted after the meetings in both study groups (EG and CG). Its course was the same as the pre-test. When analysing the results of the pre-test and post-test, the focus was on determining the colours of the painted rectangles (see Figure 4).

## 3. Results

*Pre-test*. The results of the test conducted before the children started using the educational programme represented the knowledge of the children surveyed (N = 24) about the phenomenon of additive colour. By colouring rectangles on a sheet of paper, the children marked the effect of combinations of two or three lights at one point. The analysis focused on establishing the validity of this inference. Correct answers were given by 8.3% of respondents and mostly involved a combination of red and blue light. Incorrect answers were given by 78.2% and *I don't know* answers by 13.5%. The low level of correct answers

was due to the children's lack of experience in learning about this phenomenon. The detailed distribution of responses is shown in Table 1.

**Table 1.** Distribution of responses (pre-test) of the additive colour phenomenon, separately for the experimental and control groups.

| | Response | Green-Red | Blue-Green | Red-Blue | Green-Red-Blue |
|---|---|---|---|---|---|
| **EG (12)** | Correct | Yellow (0) | Blue-cyan (0) | Violet-magenta (5) | White (1) |
| | Incorrect | In total * (2) | Red (2) | In total * (3) | In total * (3) |
| | | Orange (2) | Orange (2) | Green (2) | Violet (1) |
| | | Red (2) | Green (2) | Yellow (1) | Red (1) |
| | | Green (1) | Dark green (1) | Orange (1) | Blue (1) |
| | | Violet (1) | Dark blue (1) | | Orange (1) |
| | | Blue (1) | Black (1) | | Brown (1) |
| | | | | | Dark blue (1) |
| | | | | | Black (1) |
| | *I don't know* | 3 | 3 | 0 | 1 |
| **CG (12)** | Correct | Yellow (0) | Blue-cyan (0) | Violet-magenta (2) | White (0) |
| | Incorrect | In total * (4) | In total * (4) | In total * (4) | In total * (3) |
| | | Orange (3) | Red (3) | Green (1) | Orange (3) |
| | | Violet (2) | Yellow (2) | Brown (1) | Violet (1) |
| | | Blue (1) | Dark green (2) | Blue (1) | Brown (1) |
| | | Dark blue (1) | | Orange (1) | Yellow (1) |
| | | | | Pink (1) | |
| | *I don't know* | 1 | 1 | 1 | 3 |

\* In total—the answers of children who coloured the rectangle with two or three colours next to each other. These children chose the same colours that the foils represented (e.g., orange, purple and green), see: Figure 4.

*Children's learning about the phenomenon of additive colour using the MEP (educational intervention).* After the pre-test, the EG students began the learning phase at the computer. In pairs, they used the MEP in an isolated room for eight meetings. While using the MEP, students ran various tasks. The additive colour task was opened between three and nine times. Analysis of the recordings of the children's use of the MEP made it possible to identify childish behaviours that were differentiated by their degree of approximation to the teaching strategy adopted by the authors of the Socrates103 programme. Teaching strategies are understood as the efforts of the authors of the programme to make the experiments interesting for the students, to make them simple, easy to understand and to encourage them to make an effort to solve the tasks. In order to use the programme in the way the authors had planned, students had to demonstrate an interest in the programme, motivation to make an effort, adequate reading ability and reading comprehension level, an adequate vocabulary level (of physical phenomena), an adequate level of mental development to see the relationships between objects on the screen (e.g., torches and foils), an understanding of the test situation, and the ability to enjoy the discovery for further challenges. The children's key behaviours in terms of the purpose of the study became apparent when fascination with the programme passed and difficulties with solving tasks for points emerged. An analysis of the students' behaviour was carried out. Transcripts of the behaviour of the children using the MEP were produced and the number of inputs to the additive synthesis task was recorded. Having at their disposal a recording of each child's entry into the additive synthesis task in the MEP programme, attention was paid to gestures and facial expressions, words to the peer and the researcher, and actions expressed with the mouse cursor on the screen (including experiments performed on the screen and answers to questions marked in the programme). Based on the child's actions, inferences were made about how the phenomenon was learned. There were two specific

behaviours of the students that were qualitatively different from the others (10 students). The difference in their behaviour concerned the convergence with the teaching strategy adopted by the MEP authors.

Łukasz's behaviour was as close as possible to the teaching strategy from the beginning of the use of MEP**.** Łukasz was the only EG student who made an effort to read all the programme instructions and use the clues when solving the tasks. He opened the additive colour task a total of seven times. During the first meeting, after reading the instructions, he identified what actions he could perform on the screen and, speaking to himself, drew conclusions (e.g., *red and green gives yellow*). Initially, he combined only individual colours and then all of them together. Again, he said to himself *(they all give white, and here blue, and here pink).* He opened the additive colour task again during his third, fourth, fifth and sixth use of the computer. Each time, he read the instructions and then proceeded to mark the answers. In a moment of uncertainty, he stopped to carry out the experiment and then returned to marking the answers, being sure about the solution. The next time the tasks were opened, he performed them faster. When Łukasz undertook solving the tasks, he never marked the answer without prior reflection. Whenever he made a mistake (61 times), instead of moving on to a new task (like the others) he read the correct answer indicated by the programme. As a result, he correctly solved the most tasks of all students.

Filip's behaviour was characterised by investigation of the author's teaching strategy during subsequent meetings of the use of MEP. He opened the additive colour task the most times, i.e., nine times. During the first meeting, the boy read the instructions, tried to move the lamps and combined the three colours. He answered the questions incorrectly and again tried to figure out how to get white light. The boy read the tasks but found it difficult to understand the content. He made mistakes and the technical difficulties made him get frustrated faster. He began to mark answers at random, without thinking. He opened the additive colour task at each meeting. At the third meeting, he noted the opportunity to check the number of points obtained, and from that moment he tried to get the highest number of points. He began to read the task carefully and checked the correctness of the answers. He obtained a lot of points for correctly solved tasks. He was also motivated by the opportunity to work with other students. He exchanged information about the correctness of tasks. As early as the third meeting, he was able to give correct answers to all the tasks on additive colour. At the sixth meeting, he even stated that the task about additive colour was easy, and it was easy to "earn" points on it.

The remaining students surveyed (10 students) avoided the teaching strategy used in the MEP. In presenting the most common behaviour of students, I will cite Michał. He only respected the programme's instructions at the first meeting, and carried out the experiment by manipulating the torches. Later, however, clearly bored, he began to browse through the programme skipping tasks and stopping at the more interesting ones. Working with a peer, he overheard his efforts and copied his colleague's solution. He gave up on checking the answers and correcting mistakes, and at the last meeting Michał was clearly not interested in the programme. This behaviour was displayed by the largest number of students (10). They used the MEP as if it were a toy. As the fascination stage passed, students jumped between tasks and stopped at the ones that were more interesting. They read the tasks superficially and often, without reading the instructions, tried to solve them intuitively. The behaviour of objects on the animation in the MEP were sometimes commented on aloud. In meetings where competition was encouraged, students tried to learn the answers to the programme's tasks by heart. As a result, students in this group made a large number of errors when solving the programme's tasks. This was the behaviour of the majority of students and was similar to that of students using other educational programmes [20]. It should be added that some children, when completing the last MEP meeting, were unable to give correct answers to the task on the phenomenon of additive colour.

*Post-test.* A test administered to the experimental group after a series of meetings with the MEP expressed not only the effectiveness of the programme, but also the

children's susceptibility to learning about the phenomenon on the computer screen in the way the authors of the MEP had planned. When conducting the test, the students recognised the additive colour task. They indicated that it was the same as in the first test (pre-test), and students in the experimental group indicated that the task was similar to the MEP. The results of the post-test proved to be slightly better than those in the pre-test. Fewer students said they did not know what the effect of the light combination was (6.3%), the number of correct answers increased (9.4%) and wrong answers fell (84.3%). The detailed distribution of the responses of children in the experimental and control groups in the post-test is shown in Table 2.

**Table 2.** Distribution of responses (post-test) concerning the additive colour phenomenon, separately for the experimental and control groups.

| | Response | Green-Red | Blue-Green | Red-Blue | Green-Red-Blue |
|---|---|---|---|---|---|
| | Correct | Yellow (0) | Blue-cyan (0) | Violet-magenta (5) | White (1) |
| EG (12) | Incorrect | Yellow (0) | Blue-cyan (0) | Violet-magenta (3) | White (2) |
| | | Violet (4) | Red (3) | Green (3) | In total * (5) |
| | | Blue (4) | Orange (3) | In total * (2) | Yellow (2) |
| | | In total * (2) | In total * (2) | Yellow (2) | Violet (1) |
| | | Brown (1) | White (2) | Grey (2) | Red (1) |
| | | | Yellow (1) | | |
| | *I don't know* | 1 | 1 | 0 | 1 |
| | Correct | Yellow (0) | Blue-cyan (0) | Violet-magenta (4) | White (0) |
| CG (12) | Incorrect | Blue (3) | In total * (3) | In total * (3) | In total * (4) |
| | | In total * (3) | Red (2) | Green (4) | Black (3) |
| | | Red (2) | Yellow (2) | | Violet (1) |
| | | Violet (2) | Dark blue (2) | | Brown (1) |
| | | Orange (1) | Dark green (1) | | Yellow (1) |
| | | Pink (1) | Violet (1) | | |
| | | | Brown (1) | | |
| | *I don't know* | 0 | 0 | 1 | 2 |

* In total—the answers of children who coloured the rectangle with two or three colours next to each other. These children chose the same colours that the foils represented (e.g., orange, purple and green), see: Figure 4.

The effectiveness of learning about the phenomenon of additive colour by EG students using the MEP can be made by comparing their results with the test results of the CG students. Let me remind you that the phenomenon of additive colour is not available to children on a daily basis and that specially organised activities are needed to learn about it. Students from the EG who had the opportunity to learn about this phenomenon should show more correct answers compared to students from the CG. A comparison of the results of the two groups is shown (in Table 3). It shows the change that occurred (or not) between the pre-test and post-test in both study groups.

**Table 3.** Comparison of pre-test and post-test results in the EG and the CG.

| | No change. Both Scientific Answers | | Progress Change from an Incorrect Answer to a Scientific One | | Regress Change from a Correct Answer to an Incorrect One | | No change. Both Answers Incorrect | |
|---|---|---|---|---|---|---|---|---|
| | EG | CG | EG | CG | EG | CG | EG | CG |
| Green-red | | | | | | | 12 | 12 |
| Blue-green | | | | | | | 12 | 12 |
| Red-blue | 2 | 1 | 1 | 3 | 3 | 1 | 6 | 7 |
| RGB | | | 2 | | 1 | | 9 | 12 |

The comparative analysis shows that in the answers to the combinations of green and blue light and green and red light, all students, from both the EG and the CG, gave incorrect answers. Changes were taking place in explaining the combination of blue and red light. Łukasz and Filip (EG) and Oliwia (CG) knew the correct answer already during the pre-test and marked the same answer in the post-test (there was no change, but the scientific view was maintained). Igor (EG) and three students (CG) changed the answer from incorrect to correct (there was progress). In contrast, three students (EG) and one student (CG) changed their opinion from scientific to incorrect (regress). Incorrect answers in the pre-test and post-test were given by six students in the experimental group and seven in the control group. The explanation of combining the three lights together, on the other hand, was incorrectly shown in both tests by nine EG and all twelve CG students. Progress occurred in only two students and regression in one student from the EG.

Based on the above, it can be concluded that the lack of change in the indications of green-red and green-blue colour combinations among EG students indicates that despite having access to the programme, which created conditions for learning about this phenomenon, the programme did not affect the understanding of the phenomenon to a sufficient extent for the students to use their knowledge outside the computer (in the paper-and-pencil test). In contrast, the number of changes from incorrect to correct (progress) among students in the experimental group was similar and not necessarily due to the effectiveness of the programme. However, given the way Łukasz and Filip behaved, one can be sure that the MEP used in the study (Socrates103) was effective. However, this success should be attributed not only to the programme (the teaching strategy developed by the authors), but also to the favourable capacities of the children who made the effort to meet the teacher's strategy (their cognitive abilities, reading skills, attention and interest) and to the conditions of the environment organised by the researcher to encourage the effort of solving the tasks in the programme (e.g., noting the points scored and working in pairs).

*Children's beliefs about the effect of the additive colour process.* By analysing recordings of children's statements in front of the computer screen and their behaviour during the test (pre-test and post-test), attention was paid to statements indicating an understanding of the phenomenon of additive colour. It turned out that, apart from statements similar to the scientific ones, that resulted from previous experience, the students' incorrect statements could be divided into three concepts:

(a) The effect of additive colour is the same as subtractive colour (RGB = CMY). The children were convinced that the result of mixing coloured lights (additive colour) would be analogous to the effect of mixing paints (subtractive colour), which they knew from personal experience. Children displaying this view believed that to establish the colour combination of, for example, red and green, it was sufficient to paint a rectangle with two crayons: red and green. In this way, colouring the space with three colours (RGB) resulted in a colour resembling black.

(b) The additive colour effect of combining two colours results in a third primary colour (R + G = B). Using the colour wheel discussed at school, students believed that combining two primary colours produced a third one. For example, combining red and green will produce a third colour—blue. In the situation of combining three beams of light, these children divided a rectangle (drawn on the test sheet) into three parts and coloured each part with one colour of glowing lights (RGB). The interpretation concluded that merging does not take place. See: Figure 4.

(c) The additive colour effect is the basis for the formation of any colour In this group, students explained the merging of coloured lights by trial and error. It should be added that the statements of the children in this group were characterised by a high degree of uncertainty.

## 4. Conclusions

The aim of the research was to establish children's understanding of the phenomenon of additive colour, to determine how they learn about this phenomenon using a special computer programme, and to establish whether they can use the knowledge gained in the MEP outside of the virtual reality.

The research confirmed that only a few children were able to identify the correct effect of the additive phenomenon (and only to a small extent) before they started using the MEP. This confirms the fact that children do not have many opportunities to learn about the phenomenon of additive colour. The reason for the high number of incorrect answers given may be a misunderstanding of the basic properties of light, e.g., determining the relationship between the light source and the illuminated plane [4,8,17], and the multiple contexts of the phenomenon [2].

The analysis of the tests and children's statements confirmed previous research findings of confusion between additive and subtractive colour effects [2,4]. The richness of personal experience of mixing pigments means that children who are unsure of the conclusions try to explain the phenomenon by analogy with familiar situations of pigment mixing. Evidence of this was provided by the children's statements recorded while solving problems on the computer screen. Among the children's statements, other types of reasoning were also noticeable. Some children claimed that the effect of the additive colour phenomenon was to produce a third primary colour (e.g., R + G = B). No similar interpretations have been found in the literature to date. Probably the effect of such reasoning could have been regarded as the result of guesswork (determining the outcome by trial and error). In this study, this type of reasoning was separated from indications of any colour (e.g., random guessing).

By organising the learning situation at the computer, it was assumed that the MEP would influence students' knowledge of the phenomenon of additive colour. Previous research has suggested that there is potential for effective use of the computer in explaining the properties of light to children [17]. It was felt that a similar effect could be achieved by providing children with unlimited access to the MEP, which is a virtual experiential platform. As it turned out, only two children gained competence by improving their test score (and only in two of the four phenomenon areas). When evaluating the educational effectiveness of the Socrates103 MEP, it is important to note that, as a virtual experiential platform, it demonstrates similar activities to those proposed for teachers to perform in class with the children [3]. What proved crucial in the educational effectiveness of the programme was not so much the strategy of the programme's authors, but a complex of factors. In addition to the construction of the programme (tasks, experiments and scoring), the conditions organised by the researcher also seem to be important, such as the positioning of the computers in relation to each other and the possibility of working in pairs. These external conditions favoured Łukasz and Filip, whose personality traits allowed them to acquire knowledge of the phenomenon of additive colour.

The fact that students who managed to determine all the results of the light beam combinations on the computer screen did not manifest these results in a paper-and-pencil test demonstrates that the knowledge gained on the screen appears to be constructed at a different level of representation. Perhaps the result could have been different if tasks in both tests had been performed not by drawing, but by 3D torches. Although the students remembered the situation of combining the lamps on the screen, they could not recall the colour effect. Computer animation, although readable by the screen user, does not register strongly enough in the child's memory. So far, research with computer support has been used to supplement teaching activities [17]. In this research, the programme was used only as the focus of the intervention, and the role of the researcher was reduced to triggering activity in the children, without going into the content of the MEP. This proved effective only for individuals (2) who had strong intrinsic motivators for mental effort. This raises the question of whether the programme would have been more effective if it had complemented the classroom experience, played a role of testing the children's

existing knowledge and enabled them to test it in a new (virtual) environment. The answer to this question requires empirical confirmation.

The conclusion drawn from the behaviour of the majority of the students surveyed is that multimedia experiential platforms, without the proper support of the teacher, are treated by children as toys. To use MEPs effectively in education, the teacher needs to develop a supportive scenario, structure the discovery process and guide the children's cognitive attention accordingly [15]. The basis of this support must be knowledge of optical phenomenon and knowledge of children's abilities and limitations related to the nature of cognitive development [4,5,17].

**Funding:** This research was funded by Maria Grzegorzewska University in Warsaw, grant number BSTP 5/17-I.

**Institutional Review Board Statement:** The study was conducted in accordance with the Declaration of Helsinki, and approved by the Ethics Committee of Maria Grzegorzewska University in Warsaw (protocol code BSTP 5/17-I).

**Informed Consent Statement:** Informed consent was obtained from all subjects involved in the study.

**Data Availability Statement:** The results of the study are first reported in this article and have not been coded anywhere for data protection reasons.

**Conflicts of Interest:** The author declares no conflict of interest.

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
