# Peer review of "Exploring the Phenomenon of the Additive Colour Process While Using a Computer Programme by 7–8-Year-Old Students"

_education, doi:10.3390/educsci12110740_

Round 1

Reviewer 1 Report

- line 34: intensities or frequencies?

- lines 41-45: The authors refer to subtractive colors which appear in Figure 1b? If so, perhaps they should refer to idealized cyan and magenta and not to blue or sky-blue and violet respectively. However, if the authors want to refer in pigment mixing subtractive colors in terms of reality, then Figure 1b should display the colors they mention (blue, sky-blue, violet) or add a Figure 1c with these colors.

- line 64: Perhaps the authors should mention what are the properties of light in which learners have difficulty to understand.

- line 65: The authors state: ".....they do not distinguish between additive and subtractive colours...". That is, they do not perceptually distinguish RGB from CMY? Please explain.

- line 114: What color do the torches have during the demonstration? There were torches with the three colors R, G, B that each individually illuminated the films one by one? Or did the torches have white light and illuminate the three films sequentially? Please clarify.

Do you mean film instead of foil?

- line 116: “... What will be the colour of the light if I have two torches...”. The authors mean the color of light after passing through the film? Please clarify.

- lines 116-117: “....put blue foil under one and red foil under the other and illuminate the same spot on the table with them?...” . That is, to put one film on the other and illuminate with a white light torch? Please clarify.

- lines 118-121: The rectangle what is it supposed to be? A white sheet of paper? A Colored Film? Please clarify.

- lines 131-132: Figure 4 is the drawing of a particular child? Please state it.

- Somewhere at the beginning of section 2 the research design should be explicitly mentioned, i.e. that there was a pre-test based on students' drawings, a teaching intervention based on the software MEP and a post-test based on....(I have missed this. Was the posttest like the pretest or was it based on what is contained in Figure 2?). Ok in lines 180-181 the authors report that the posttest was the same as the pretest. However, the full research design should be explicitly mentioned at the beginning of section 2.

- lines 215-216: How was the analysis done?

- line 218-...: It should initially be reported that 3 behavioral models with the corresponding frequencies were detected.

- line 258-....: Does everything mentioned in this paragraph apply to the third model or is it another category?

- 390-....: Perhaps it should be emphasized that the results might have been different if the tasks in the two tests were not done by drawing but by 3-D torches.

Author Response

- line 34: intensities or frequencies?

Thank you very much for this comment. Considering the structure of the displays, they are composed of an array of diodes with three/four colours (RGB+G). A beam of these colours is called a pixel. Colour variation is produced by intensifying one colour and blanking out another. For example, when producing a white colour, the intensity of all three/four colours must be increased, a black colour must be intensified by extinguishing all three/four colours, a green colour is the result of lowering the intensity of green and blue, and so on. Based on the above, we speak of intensities.

- lines 41-45: The authors refer to subtractive colors which appear in Figure 1b? If so, perhaps they should refer to idealized cyan and magenta and not to blue or sky-blue and violet respectively. However, if the authors want to refer in pigment mixing subtractive colors in terms of reality, then Figure 1b should display the colors they mention (blue, sky-blue, violet) or add a Figure 1c with these colors.

Thank you very much for your feedback. The colour image has been replaced with an in-house development that takes into account colours from the hexadecimal system (e.g. yellow FFFF00). The inclusion of this system ensures the correct colour scheme in the illustration.

- line 64: Perhaps the authors should mention what are the properties of light in which learners have difficulty to understand.
- line 65: The authors state: ".....they do not distinguish between additive and subtractive colours...". That is, they do not perceptually distinguish RGB from CMY? Please explain.

Thank you very much for your remark. The highlighted fragment has been corrected.

- line 114: What color do the torches have during the demonstration? There were torches with the three colors R, G, B that each individually illuminated the films one by one? Or did the torches have white light and illuminate the three films sequentially? Please clarify. Do you mean film instead of foil?

Thank you very much for your question. I use three foils, each in a different colour, and a torch with a white light. A fragment of the text has been supplemented

- line 116: “... What will be the colour of the light if I have two torches...”. The authors mean the color of light after passing through the film? Please clarify.

Thank you very much for your question. Yes, it is about the light passing through the foil.

- lines 116-117: “....put blue foil under one and red foil under the other and illuminate the same spot on the table with them?...” . That is, to put one film on the other and illuminate with a white light torch? Please clarify.

Thank you very much for this question. It is about two separate torches, one foil is to be connected to each. The text has been updated.

- lines 118-121: The rectangle what is it supposed to be? A white sheet of paper? A Colored Film? Please clarify.

Thank you very much for your question. The rectangle on the sheet of paper is white (the background of the sheet of paper). The children colour the rectangles themselves with a crayon in whatever colour they think is appropriate. Corrected in the text.

- lines 131-132: Figure 4 is the drawing of a particular child? Please state it.

Thank you very much for your question. Signature completed.

- Somewhere at the beginning of section 2 the research design should be explicitly mentioned, i.e. that there was a pre-test based on students' drawings, a teaching intervention based on the software MEP and a post-test based on....(I have missed this. Was the posttest like the pretest or was it based on what is contained in Figure 2?). Ok in lines 180-181 the authors report that the posttest was the same as the pretest. However, the full research design should be explicitly mentioned at the beginning of section 2.

Thank you very much for your valuable comment. The text has been updated. 

- lines 215-216: How was the analysis done?

Thank you very much for your question. The text has been completed. 

- line 218-...: It should initially be reported that 3 behavioral models with the corresponding frequencies were detected.

Thank you very much for your valuable comment. The text has been corrected.

- line 258-....: Does everything mentioned in this paragraph apply to the third model or is it another category?

Thank you very much for your question. Yes, the whole paragraph is about the third behaviour. The text has been amended.

- 390-....: Perhaps it should be emphasized that the results might have been different if the tasks in the two tests were not done by drawing but by 3-D torches.

Thank you very much for your valuable comment. The text has been updated.

Reviewer 2 Report

Dear authors, please find my comments on the following attachment

Author Response

The article is written on a very interesting topic, that of exploring the phenomenon of additive colour process through using a computer-based multimedia educational program (MEP). While the authors have made a remarkable effort, there are some issues in the article that needs to be addressed. These are presented in the following bullets points In the text there seems to be a great inconsistency between the research questions and the results section. The authors should made clear how their data ‘answers’ to each research question

- You should provide a literature review on the behavioral models referred on lines 215-217

- You should clarify with which criteria each student was ascribed to each behavioural model. Moreover, it would be useful to provide, maybe in a table, the percentage of students that ascribed to each behavioural model

Thank you very much for this valuable comment. When elaborating on students' behaviour, I actually took into account the similarity of behaviour. So I did not use any theories, and for this reason - to create illusions - I removed the term behavioural model and concentrated on identifying two behaviours of children with a specific way of using MEP.

- You should give more information to the reader on the paper-and-pencil test that you refer on line 391. Given the fact that this test is related with the 3rd research question of the study, it is important to be presented on the results part.

Thank you very much for this comment. The entire test procedure (including the paper-pencil sheet) is described in lines 109-127. For clarification, the text has been amended.

-In lines 325-348 the children’s beliefs about the effect of the additive colour process are presented. Please make more explicitly the relation of this part with your 1st research question. You may also need to rephrase this research question

Thank you very much for this comment. Indeed the wording was not precise. The text has been corrected (first research objective).

- Your 2nd research questions need also to be addressed. It is not clear to me whether you really explored the ‘how they learn about the phenomenon of …’

Many thanks for this valuable comment. Edited the text by removing the second research objective.

- Please add a paragraph with the study limitations - You should clarify the meaning of the sentences in the following lines

  • Lines 22-24
  • Lines 39-40

Thank you very much for your attention. Text amended

- On lines 47-48 please write the letter first (a) & (b) and then the text. That is: (a) the blending of light colours in the phenomenon of additive colour (b) The blending of paint pigments in subtractive colour

Thank you very much for your feedback. The text has been updated.

- On line 90 the indicative letter (a) is missing

Thank you very much. The text has been completed